# Additive Manufacturing of Cementitious Materials by Selective Paste Intrusion: Numerical Modeling of the Flow Using a 2D Axisymmetric Phase Field Method

**DOI:** 10.3390/ma13215024

**Published:** 2020-11-07

**Authors:** Alexandre Pierre, Daniel Weger, Arnaud Perrot, Dirk Lowke

**Affiliations:** 1L2MGC EA4114, CY Cergy Paris University, 5 mail Gay-Lussac—Neuville-sur-Oise, 95031 Cergy-Pontoise, France; alexandre.pierre@cyu.fr; 2Centre for Building Materials (CBM), Technical University of Munich, Baumbachstraße 7, 81245 Munich, Germany; weger@ib-schiessl.de; 3IRDL, UMR CNRS 6027, Université Bretagne Sud, F-56100 Lorient, France; 4Institute of Building Materials, Concrete Construction and Fire Safety (iBMB), University of Braunschweig, Beethovenstr. 52, 38106 Braunschweig, Germany; D.Lowke@ibmb.tu-bs.de

**Keywords:** 3D printing, selective paste intrusion, yield stress, cement, numerical simulation

## Abstract

The 3D printing of concrete has now entered a new era and a transformation of the construction sector is expected to reshape fabrication with concrete. This work focuses on the selective paste intrusion method, which consists of bonding dry particles of aggregate with a cement paste. This innovative technique could lead to the production of very precise component for specific applications. The main obstacle to tackle in order to reach a high shape accuracy of high mechanical performances of 3D printing elements by selectively activating the material is the control of the distribution of the cement paste through the particle bed. With the aim to better understand the path followed by the solution as it penetrates a cut-section of the granular packing, two-dimensional numerical modeling is carried out using Comsol software. A phase-field method combined with a continuous visco-plastic model has been used to study the influence of the average grain diameter, the contact angle, and the rheological properties of cement pastes on the penetration depth. We compare the numerical modeling results to existing experimental results from 3D experiments and a one-dimensional analytical model. We then highlight that the proposed numerical approach is reliable to predict the final penetration of the cement pastes.

## 1. Introduction

Digital construction using the 3D printing of concrete has been democratized in many countries around the world as large scale projects have emerged and scientific knowledge have grown rapidly in the last few years [1,2]. The free-form possibility allowed by 3D-printers is one of the opportunities among other expected advantages, but several challenges regarding both materials and processes are remain in the digital concrete industry [3,4].

The most common method is the deposit of the cementitious material using digitally controlled extrusion process [4,5,6,7,8,9,10]. The basic principle is to combine a well-defined and precise process using robots with a material for which the rheological behaviour and structural build-up are well characterized. For extrusion-deposition techniques, the material must be fluid enough to be extruded from the nozzle, but the success of the printing process mainly depends on the fine-tuning of the construction rate according to the material structural build-up [9]. The rheological requirements for printable concrete deal with several physical properties, such as yield stress, thixotropic effect, critical strain, or elastic modulus, as reported in [11]. In addition to the social, economic, and durability requirements, the scientific challenge is huge and recent work shows that the rheological requirements of the cement based materials used are dependent on both the process and object shape [11].

An alternative is to use an additive layer-wise process based on the a selectively activated powder-bed by injecting a fluid or an activator [2,4]. After successive depositions of thin layers of powder, complex shapes and overhang structures can be produced by selectively depositing the fluid activator which ensures the particles binding at the hardened state. Unbounded particles are then removed and could be used for other applications or even 3D printing [2]. Finally, a post-print treatment could be applied to improve the final properties of the structure regarding to thermal or mechanical performances [2].

In principle, particle-bed 3D printing methods can be divided in selective binder activation, selective paste intrusion and binder jetting [2]. Depending on the application, selective binder activation can be used: a mixture of aggregate and binder is poured and the hardening will be activated by spraying or jetting of a solution of water and admixture [2,12,13,14,15]. Recent works [2,16,17,18] used the selective paste intrusion method to build cement based material components: the binder composed of water, cement, and admixtures is applied to a bed of sand particles. Considering the binder jetting, the binder can be a resin which reacts with a constituent of the particle bed, but this last technique is less used for civil engineering applications nowadays.

Advantages of the particle-bed 3D printing techniques are that the final shape of the element or the structure only depends on the injection and the penetration of the binder as the sand particle bed is mechanically stable at rest. Furthermore, cantilevers or other overhanging elements can be built without the need for additional support structures due to the presence of unbound particles. This technique aims to be resource efficient compared to the extrusion method as fewer polymer additives are needed to produce complex structural elements. The selective paste intrusion method has already allowed to produce high strength concrete parts [2,16,17,18]. Nevertheless, the selective paste intrusion has not yet been fully transferred to the concrete construction industry. However, good material properties depend on a sufficient penetration of the layers. As the future strength and final shape accuracy is related to the degree of paste penetration, it is a major concern for the additive manufacturing of cementitious materials. Therefore, the penetration of the binder should be predicted depending on several factors, such as the rheological properties of the liquid binder, the particle-bed morphology, and the interface properties between fluid (cement suspensions) and solid (particles). A way to optimize the process could be the application of numerical modeling to predict quantitative results of the final penetration depth.

The numerical modeling of fresh concrete and cementitious material flows has gained interest in the last decade and a special technical letter concerning the new insight and challenges about this subject has been recently published [19]. For 3D printing with particle-based methods, the challenge is to capture the evolution of free surface with time as the moving interfaces of two-phase flows to capture the fluid flowing through air content of the porosity of the particle-bed. Three different approaches can be used: interface tracking where materials flows through a fixed mesh (Eulerian), interface capturing where the motion of materials particle is tracked (Lagrangian), or ALE (arbitrary Lagragian–Eulerian), which combines both but where free surface movement is limited [20]. In Eulerian tracking methods, an additional field variable is used to describe the interface between two fluids or a solid and a fluid. As reported by Mokbel et al. [21], several interface capturing approach can be used: level-set [22], volume of fluid [23], and phase field method [24]. Volume-of-fluid (VOF) could have been used, but it can suffer from inaccurate computations of the interface properties as reported in [25]. The VOF method is then less accurate for 3D printing with selective paste intrusion where surface tension could be the dominant effect. Also, the phase-field model is a front capturing method where a field variable is used to indicate each phases of both fluids while dealing with diphasic 3D printing process. This method includes the mass conservation and transport stabilization, which is not the case while dealing with level-set methods or other Eulerian methods. Consequently, phase field methods allow for coupling interface advection and flow equations in order to prevent time step restrictions of the used solver.

It is commonly admitted that cement pastes are viscoplastic fluids which behave as Bingham fluid. Its flow through sand particle layer depends on its rheological properties: yield stress, viscosity, the morphological properties of the sand particle-bed (layer height, medium grain diameter, shape, roughness, void volume fraction), the sand wettability, and absorption properties. Theoretical work validated by experimental results established the general relation between pressure drop as a function of the penetration rate for a yield stress fluid flowing through a porous medium [26]. This physical approach has recently been adapted to develop and validate an analytical model to evaluate the penetration of cement paste through sand particle-beds assuming no-slip between aggregates and cement paste for selective cement paste activation [27]. In addition, a recent analytical and digital modeling study [28] presents four scenarios that may arise when a drop of Newtonian fluid is deposited into a particle bed: (1) the drop does not bond the grains due to a lack of penetration, (2) the drop remains trapped between the grains, (3) the drop descends to the bottom of the layer, and finally, (4) the drop splits and spreads around the particles. These four scenarios can occur for non-Newtonian fluid like cement paste until the pressure applied to the fluid overcomes the magnitude of its yield stress. Within the above frame, the influence of the rheological properties that affect the penetration of the binder through a sand layer bed will be discussed in this study. In addition, special attention will be paid to the influence of the relationship between surface tension and the forces of gravity.

Then, the aim is to provide new information about the influence of surface tension effect and contact angle on the path followed by the solution as it penetrates the granular matrix. It is necessary to ask questions about the parameters to be controlled to produce homogeneous structures. In this study, we first present the Fourier–Voronoi-based generation from the work of Mollon et al. [29] to generate three sand bed packings which differ in their the average particle diameter and which have been used previously in the experimental measurements of Pierre et al. [27]. Then, the phase field method and the modeling of the rheological behaviour following a continuous visco-plastic model are presented to follow the path of the suspensions through the powder-bed. It is shown that the contact angle at the fluid–particle interface, the average grain diameter, and the rheological properties of the suspensions govern the kinetic of the flow and the final penetration depth. We then highlight that the numerical modeling is able to provide a quantitative estimation of the penetration final depth. A comparison of the final penetration depth with existing 3D experimental results and a 1D analytical approach is also carried out.

## 2. Materials and Methods

### 2.1. Sand Packing Generation

In this work, a Fourier–Voronoi-based generation is used to generate the sand packing in 2D following the open-source program of Mollon and Zhao [29]. Using a Matlab code, three sand packings layers of 12.5 mm width and 20 mm height were generated to model the experimental conditions of the measurements performed in Pierre et al. [27]. The void fraction of sand packings, ie. the ratio of the volume of the air phase occupied in the aggregate layer, was experimentally determined to a value of 0.46 by Pierre et al. [27]. The results of the three sand packing layers obtained are shown in Figure 1 for the average particle diameter, noted d_agg_, of 1.0 mm, 1.6 mm, and 2.6 mm, respectively.

At first sight, the 2D formulation of the proposed method may seemingly limit its practical use for real 3D particle modeling [29]. In order to qualitatively compare theses modeling results to real case, the numerical assembly in two-dimensions is considered as a cut-section of a 3D granular packing, as illustrated in Figure 2. It can be observed that, both in real (Figure 2) and in the modeling cases (Figure 1), most of the sand grains do not have an interconnection with each other. Also, the void fraction is constant for all the sand packings generated as it is computed by the code of Mollon and Zhao [29]. Then, the schematic sand packing used and schematized in Figure 1 will be used to model the flow path of the cement paste.

### 2.2. Modeling of the Rheological Behaviour

A Papanastasiou model [30] is used to describe the Bingham rheological behaviour of the cement pastes as already proposed in a previous study from the authors using a level-set method [31]. The review of Mitsoulis [32] showed that the regularized continuous visco-plastic model of Papanastasiou [30] is reliable to approximate the Bingham model. In this model, the shear rate is linearly linked to the shear stress through the dynamic viscosity of the penetrating fluid that is expressed following Equation (1):(1)$$\eta_{paste}=\eta_{exp}{\left|\dot{\gamma }\right|}^{n_{exp}-1}+\frac{\tau_{0,exp}}{\left|\dot{\gamma }\right|}\left[1-e^{\left(-m\left|\dot{\gamma }\right|\right)}\right]$$

The value of the exponent *n_exp_* is 1 as the suspensions of cementitious materials behave as a Bingham fluid [11,33,34] ($\tau =\tau_{0,exp}+\eta_{exp}\dot{\gamma })$ and also showed a reliable fitting with the rheological measurements used in this study and carried out by Pierre et al. [27]. Values of experimental yield stress $\tau_{0,exp}$ and viscosity $\eta_{exp}$ are evaluated from the fitting of the Bingham model on the measured data from rheological measurements on two different cement pastes carried out in a previous work of Pierre et al. [27] are used in this study and can be seen in Figure 3.

The shear stress–shear rates flow curves for the Bingham model obtained from experimental measurements of Pierre et al. [27] are plotted with the best fitting curve of the Papanastasiou model. It can be noted that the Papanastasiou model perfectly fits the Bingham rheological behaviour for shear rates higher than 0.01 s^−1^ with a value of m parameter of 500 s. The value of m parameter is a fitting parameter chosen to adjust the reliable fitting domain [30]. This condition was always fulfilled to evaluate reliable penetration depth for all the numerical simulations as shown by Pierre et al. [31].

### 2.3. Numerical Simulation Procedure

The numerical simulations were carried out using Comsol^®^ software and the phase field method in a 2D axi-symmetric condition. The phase-field model in the conservative form used in this study breaks the Cahn–Hilliard equation into two second-order partial differential equations [35,36]:(2)$$\frac{\partial \varphi }{\partial t}+u\nabla .\varphi =\nabla .\frac{\gamma \lambda }{\varepsilon^{2}}\nabla \psi$$
(3)$$\psi =-\nabla .\varepsilon^{2}\nabla \varphi +\left(\varphi^{2}-1\right)\varphi )$$
where φ denotes the phase field variable, ψ an auxiliary variable to separate fourth order equations into two second order equations, t the time, and ε the parameter determining the thickness of the interface; γ is the mobility (m^3^·s/kg) which determines the relaxation time of interface and timescale of diffusion. This parameter γ is determined by the relation $\gamma =\chi \varepsilon^{2}$ where χ is a mobility tuning parameter fixed to a default value of 1 m·s/kg; λ is the mixing energy quantity (N), and u is the fluid velocity (m/s).

The maximum mesh element will be taken as the incertitude while evaluating the penetration depth of yield stress fluid through sand packing. The mobility and interface thickness parameters are related to the surface tension coefficient σ (N/m) and defined through Equation (4):(4)$$\sigma =\frac{\sqrt{8}}{3}\frac{\lambda }{\varepsilon }$$

The phase field method finally solves: the Cahn–Hiliard Equation (2), a continuity equation Equation (5), and the Navier–Stokes Equation (6).
(5)∇.u=0
(6)$$\rho \left(\frac{\partial u}{\partial t}+\left(u.\nabla \right)u\right)=-\nabla P+\nabla .\left(\eta \left(\nabla u+{\left(\nabla u\right)}^{T}\right)\right)+\rho g$$

From Equation (6), it is shown that the unique driving force for the penetration is the gravity, because no initial velocity, i.e., u = 0, is considered in this study. The mass conservation is checked for each simulation in this paper as it could be a problem to ensure a careful implementation of the numerical modeling. The density and the dynamic viscosity were computed using the following equations:(7)$$\rho =\rho_{air}+\left(\rho_{paste}-\rho_{air}\right)\varphi$$
(8)$$\eta =\eta_{air}+\left(\eta_{paste}-\eta_{air}\right)\varphi$$

Figure 4 illustrates the different domains (air in blue and cement paste yield stress fluid in red) and the meshing used for the simulations at the initial time of the numerical simulations.

### 2.4. Contact Angle and Boundary Conditions

Pierre et al. [27] showed from experimental results that the water content of the sand affects the penetration depth during 3D printing process like the selective paste intrusion method. Sand particles absorb some of the water from the cement paste leading to an evolution of the yield stress and the viscosity during its penetration through the particle-bed. The effect of the surface wetting and absorption of water by dry particles of sand on fluid penetration could be modelled with different boundary conditions at the sand-percolating fluid interface. Considering the numerical simulations, a way to study the fluid-solid contact influence on the penetration is to analyze the influence of the contact angle at the interface on the penetration. The contact angle values studied ranged from 3π/8 to 5π/8 (see Figure 5). At the domain boundaries, the contact angle was kept constant throughout all simulations at a value of 90° to avoid the fluid being attracted to the walls as proposed by Boyce et al. [28]. Also, the no-slip condition is defined as a fluid velocity which tends to 0 at the fluid-solid interface.

The slip length is the height of the element size taken between 0.025 mm and 0.563 mm depending of the medium diameter of the sand bed particles. The discretization scheme, the solver settings and the time step width are standard setting of Comsol^®^ software. Concerning the boundary conditions, no mass flows across boundaries is allowed.

## 3. Results

### 3.1. Influence of the Contact Angle

Once the yield stress is overcome, the motion of the cement paste through the granular matrix is governed by two properties: a term due to gravity and a term due to the surface tension as well as the contact angle. This part aims to define the influence of the contact angle on the penetration of the cement paste through the sand particle beds of different average diameter. Snapshots of the penetration of the fluid having a yield stress of 2 Pa illustrate, in Figure 6, the temporal development of the cement paste distribution through the granular beds depending of the value of the contact angle for a value of d_agg_ = 1.6 mm. These snapshots are from a revolution of the 2D axisymmetric numerical modeling.

The influence of the values of the contact angle is shown in more details as a function of time in Figure 7 and Figure 8 plotting the penetration depth of respectively pastes of a value of yield stresses of 2 Pa and 4 Pa through a granular packing of d_agg_ = 1.6 mm. A value of d_agg_ = 1.6 mm of particles was chosen to study the influence of the contact angle as the ratio $\sqrt{\sigma /\rho g}\;$is approximatively equal to 1.6 mm. 

It is observed from Figure 7 and Figure 8 that the penetration depth kinetic depends on the contact angle. A small contact angle tends to slow down the penetration depth as function of the time for both values of yield stress. On the contrary, the liquid falls to the bottom of the granular bed in a shorter time for a value of contact angle of 5π/8. In addition to the value of yield stress, the value of the contact angle at the interface between fluid and aggregates is a key parameter for the modeling of the 3D printing selective binding technique.

These results can be linked as the capillary force F=σcos(θ) vanishes if the contact angle is equal to π/2; is negative if the contact angle is 3π/8 and is positive for a contact angle value of 5π/8. Then, the contact angle with a value of 3π/8 tends to pull upward the paste and on the contrary a value of contact angle of 5π/8 tends to affect the motion by pulling downward the paste. If the value of contact angle is approximatively equal to π/2, the surface tension less affect motion as the force *d*u to surface tension and gravity are balanced and the penetration is only dependent of the rheological parameters of the fluid (yield stress and viscosity). Nevertheless, as the porous media does present surfaces which are not parallel, the resulting interface could be curved and hence result in a net pressure drop. Nevertheless, as we want to compare the numerical modeling results with previous existing results taking into account the rheological properties as well as the morphology and granular distribution of the granular bed, we choose a value of the contact angle of π/2. Within this angle value, we avoid the fluid being attracted to the grains as proposed by Boyce et al. [28].

### 3.2. Penetration Depth

Figure 9 and Figure 10 show the penetration depth as a function of time for medium diameters of the sand bed particles and the cement pastes presenting yield stresses of 2 Pa and 4 Pa respective and viscosities values of 0.1 Pa s Herein, no scaling is applied for the model as these values refer to previous experimental measurements of Pierre et al. [27]. Results from Figure 9 show that the penetration of the fluid of $\tau_{0,\exp}$ = 2 Pa is complete for medium diameters of sand bed particles of 1.6 mm and 2.6 mm. Here, the numerical simulation for the average grain diameter of 2.6 mm did not last 20 s as we introduce a stop criterion when the mass conservation loss becomes higher than 5%. The kinetic of the penetration is slower for the fluid penetrating into the sand bed of d_agg_ = 1.6 mm than the sand bed of d_agg_ = 2.6 mm. The entire penetration through the sand bed of average diameter of 1 mm is not achieved as it tends to a plateau value of 6 mm. As already shown using a level-set method [31], these results confirm that, if the contact angle is neutral, a balance between the average grain diameter of the sand particles and the rheological properties of the cement paste is crucial to achieve a complete penetration using a 3D printing particle based method process, as previously shown with the establishment of modified Darcy’s law for yield stress [36].

Figure 10 illustrates the penetration depth of the suspension having a yield stress of 4 Pa through the three different sand particles bed. We observe that plateau values occur for the sand bed particles of average grain diameters of 1 and 1.6 mm. The penetration is entirely achieved through a sand bed particle of average grain diameter of 2.6 mm.

### 3.3. Comparison of Experimental Results and Numerical Simulations

Figure 11 and Figure 12 compare the penetration depths through particle-beds of different medium diameters obtained by numerical simulations with the analytical model and experimental measurements from Pierre et al. [27]. We plot Figure 11 and Figure 12 the results of these experiments which were carried out on both dry and wetted sand with a water content of 0.6% in order to assess a qualitative comparison of the 2D fluid invasion with two different cases of real 3D experiments. We should recall here that dry sand can lead to shear-thickening of the penetrating fluid and then lower penetration values than saturated wetted sand because of water absorption and surface wetting.

The analytical model of Pierre et al. [27] allows for evaluating the height of penetration in function of paste rheological parameters and density, particles bed average grain diameter and void ratio, solid volume fraction, following Equation (9)
(9)$$h_{pen}=\frac{\rho gd_{agg}\left(1-\Phi_{s}\right)h_{lay}}{\alpha \tau_{0,exp}-\rho gd_{agg}\Phi_{s}+\frac{\Phi_{s}}{1-\Phi_{s}}6\tau_{0,exp}\kappa }$$
where *h_lay_* is the height of the layer of the sand packing bed, Φ_s_ is the sand volume fraction, $\tau_{0,exp}$ is the yield stress, *κ* is a parameter that represents the fraction of the sand particles surface area where the fluid is sheared. The coefficient *α* is a fitting parameter computed from experiments on Carbopol gels and is independent on the fluid properties. Chevalier et al. [36] propose *α* = 5.5 for a spherical particle assembly. The reader can refer to the work of Pierre et al. [27] for more details on Equation (9).

Figure 11 and Figure 12 aim to compare the results from three different approaches: the 2D axi-symmetric numerical modeling presented in this study, the 1D analytical approach of Equation (9) and the 3D experimental results presented in the work of Pierre et al. [27]. Nevertheless, it is observed from both Figure 11 and Figure 12 that the numerical modeling results show a quantitative evolution which follows the experimental measurements carried out on wet sand.

The penetration evaluated from Equation (9) shows higher penetration depth for a fluid with a value of yield stress of 2 Pa whereas a better agreement with the analytical model is found with a value of yield stress of 4 Pa. As a reminder, the experimental results with wet sand showed higher penetration than the dry sand as the water saturation of wet sand the particles prevent the absorption of the cement pastes. At a layer particle bed having an average grain diameter of 1 mm, a gap is observed between all the approaches. This comparison clearly shows that some interactions at the fluid-solid interface are not captured by both analytical 1D approach and developed 2D axi-symmetric numerical modeling for fine grains where interface between fluid and solid may govern the flow path. The gap between the experimental results and numerical simulations always ranges from one size of grains to two sizes of grains. For the sand packing with an average grain diameter of 1.6 mm, the difference of penetration depth values is about 3 mm for yield stress suspension of 4 Pa, which corresponds to two grains. Of course, the value of the penetration can be tailored with the values of the contact angles which govern the flow behaviour of the suspension. It then allows improving the penetration during the printing process through the use of surface tension modifiers that will improve the wettability. It can be noted that numerical modeling can capture the effect on heterogeneous granular packing that may create preferential flow path that cannot be predicted using the one-dimensional analytical modeling of Pierre et al. [27].

## 4. Conclusions

In this paper, we have discussed the influence of the average diameter of the sand particles, the rheological properties of fluids, and the contact angle on the influence of the penetration of yield stress fluids through a sand particles layer using numerical simulations. We emphasized that the contact angle can govern the kinetic of the penetration and the final depth penetration as reported from previous experimental works. Regarding the numerical modeling, we have shown that using a Fourier–Voronoi based generation to model a sand particle-bed and a phase-field method allows following the penetration depth of yield stress fluid through a sand particle bed. We highlighted that the contact angle is predominant in terms of the fluid behaviour traveling down the packing bed. Different scenarios can occur regarding the effect of surface tension, which is dependent on the contact angle: the penetration is accelerated, trapped, or does not infiltrate the sand bed. Also, the numerical simulations results were compared with analytical modeling, taking into account the sand particle bed properties and the rheology of the fluid and experimental results. The numerical results have provided a reliable quantitative prediction of the final depth penetration and more information about the flow path in comparison to previous analytical unidirectional modeling of the flow.

Advances in calculation clusters are expected to improve three-dimensional digital modeling by taking into account the surface tension effects and the modification of the contact angle by additives. Analytical modeling must also make advances in predicting the penetration of fluids into particles, whether through the activation of cement or the intrusion of paste.

## Figures and Tables

**Figure 1 materials-13-05024-f001:**
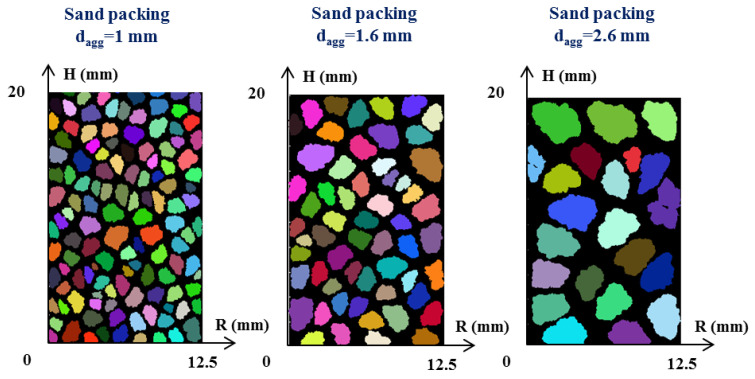
Sand packing (Φ_p_ = 0.46) generated with a Fourier–Voronoi-based generation from the program of Mollon et al. [29].

**Figure 2 materials-13-05024-f002:**
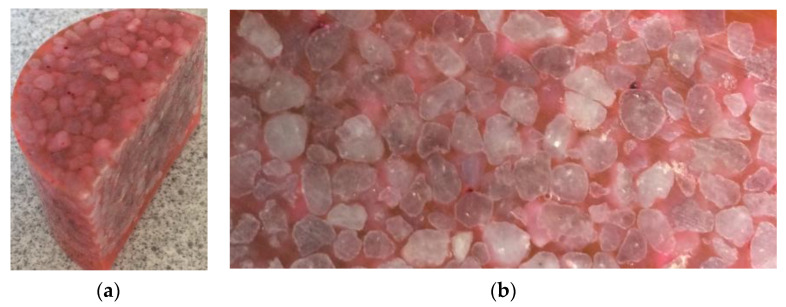
(**a**) Picture of the sand packing in 3D; (**b**) Picture of the 2D cut-section of the sand packing assembly.

**Figure 3 materials-13-05024-f003:**
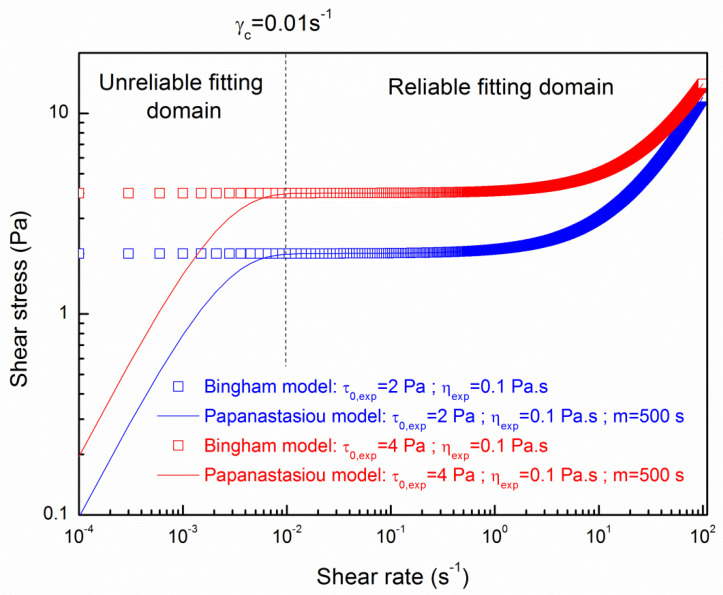
Rheological behaviour of the cement pastes modeled by the Bingham model from experimental measurements of Pierre et al. [27] approached with the Papanastasiou model.

**Figure 4 materials-13-05024-f004:**
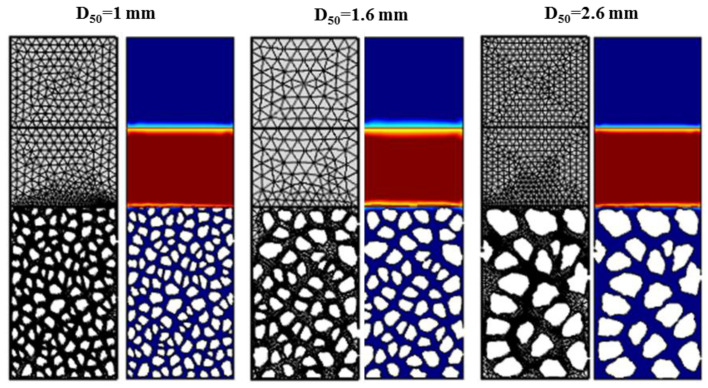
Domains where cement paste is in red and air in blue and meshing used for the numerical simulations using CFD module from Comsol^®^ software.

**Figure 5 materials-13-05024-f005:**
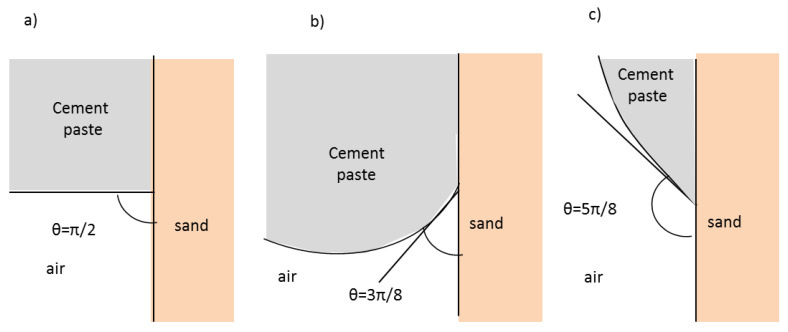
Contact angle at the interface cement-air-sand: (**a**) θ = π/2; (**b**) θ = 3π/8; (**c**) θ = 5π/8.

**Figure 6 materials-13-05024-f006:**
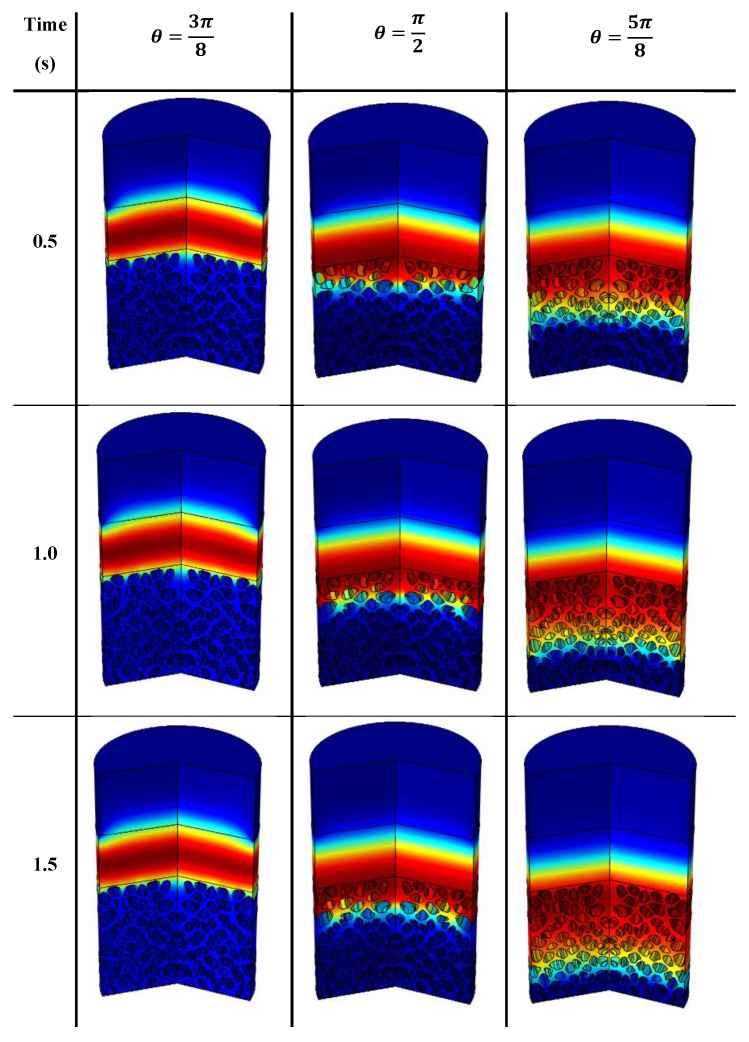
Snapshots of the penetration of a cement paste through a sand bed with particles of d_agg_ = 1.6 mm: influence of the value of the contact angle.

**Figure 7 materials-13-05024-f007:**
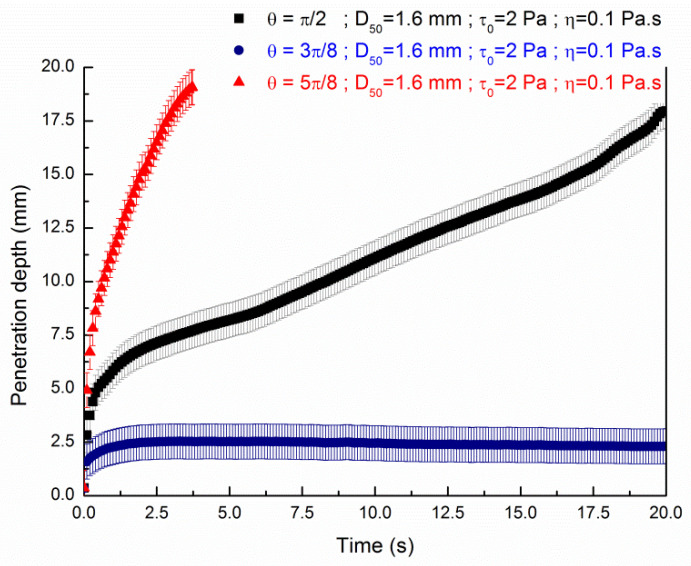
Influence of the contact angle at the interface for the penetration of a yield stress fluid $\left(\tau_{0,exp}=2\;\mathrm{Pa};\;\eta_{exp}=0.1\;\mathrm{Pa}\cdot s\right)$ in a sand bed particles of 20 mm height and d_agg_ = 1.6 mm.

**Figure 8 materials-13-05024-f008:**
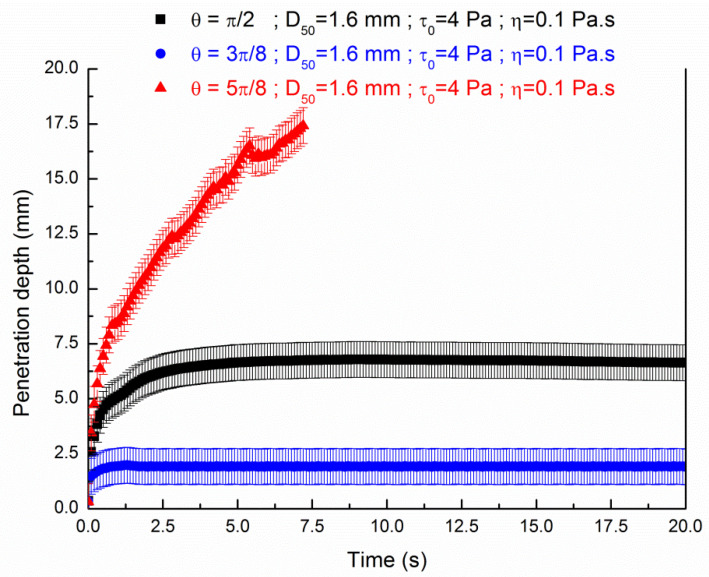
Influence of the contact angle at the interface for the penetration of a yield stress fluid $\left(\tau_{0,exp}=4\;\mathrm{Pa};\;\eta_{exp}=0.1\;\mathrm{Pa}\cdot s\right)$ in a sand bed particles of 20 mm height and d_agg_ = 1.6 mm.

**Figure 9 materials-13-05024-f009:**
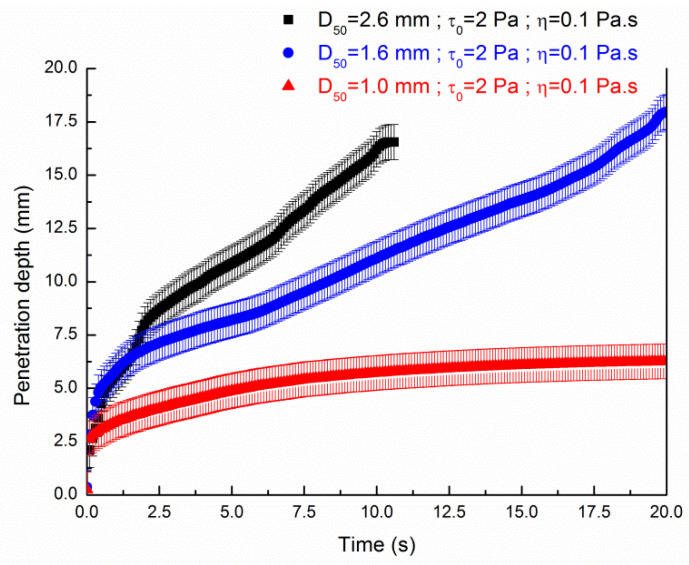
Influence of the average diameter of the sand bed particles on the penetration depth of a cement paste ($\tau_{0,exp}=2\;\mathrm{Pa};\;\eta_{exp}=0.1\;\mathrm{Pa}\cdot s)$ of 20 mm height as a function of time.

**Figure 10 materials-13-05024-f010:**
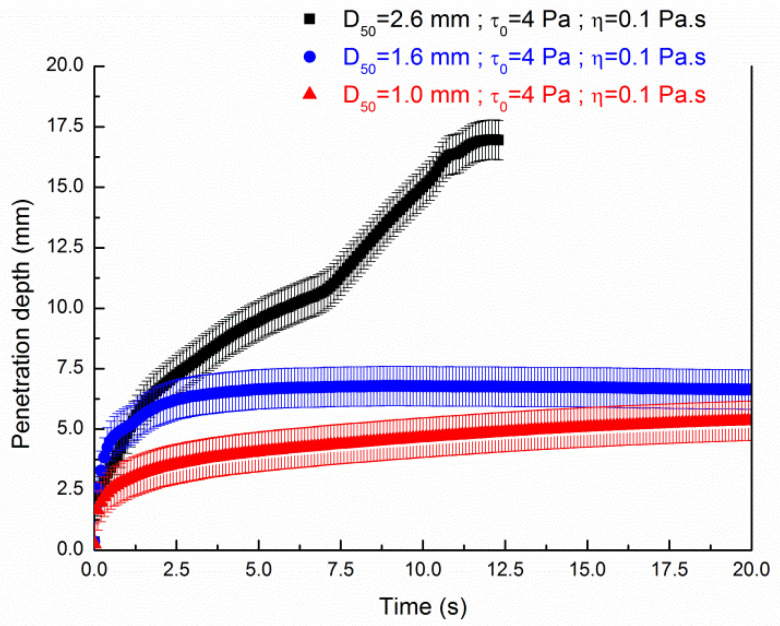
Influence of the average diameter of the sand bed particles on the penetration depth of a cement paste ($\tau_{0,exp}=4\;\mathrm{Pa};\;\eta_{exp}=0.1\;\mathrm{Pa}\cdot s)$ of 20 mm height as a function of time.

**Figure 11 materials-13-05024-f011:**
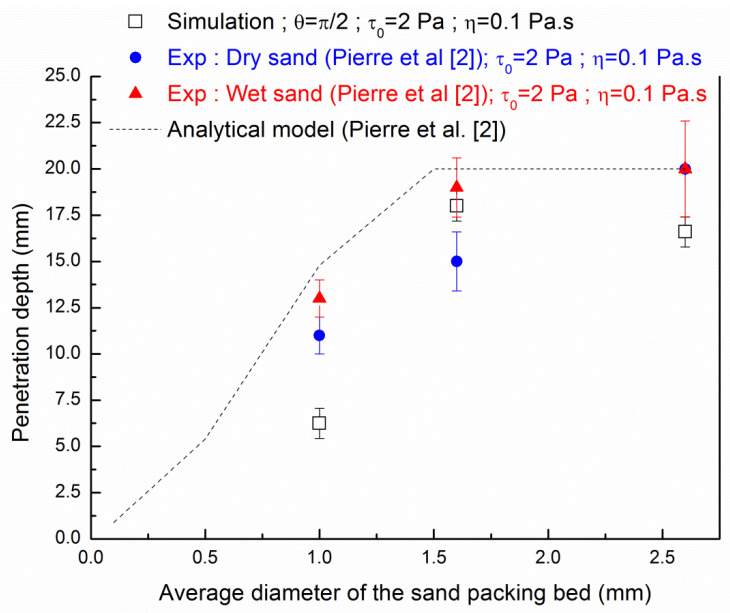
Comparison of the numerical penetration depth of cement paste $\tau_{0,exp}=2\;\mathrm{Pa};\;\eta_{exp}=0.1\;\mathrm{Pa}\cdot s)$ and the experimental results and analytical model from Pierre et al. [27].

**Figure 12 materials-13-05024-f012:**
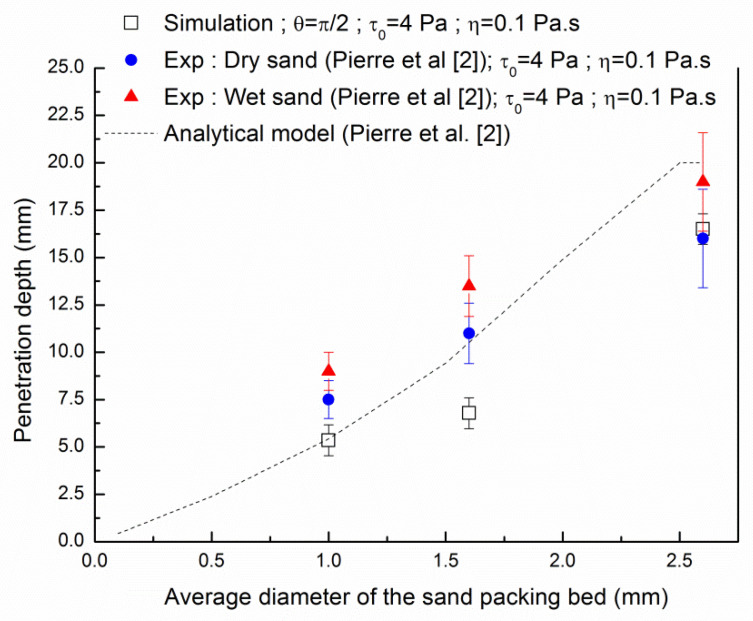
Comparison of the numerical penetration depth of cement paste ($\tau_{0,exp}=4\;\mathrm{Pa};\;\eta_{exp}=0.1\;\mathrm{Pa}\cdot s)$ and the experimental results and analytical model from Pierre et al. [27].

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
