# Peer review of "Additive Manufacturing of Cementitious Materials by Selective Paste Intrusion: Numerical Modeling of the Flow Using a 2D Axisymmetric Phase Field Method"

_materials, 2020, doi:10.3390/ma13215024_

Round 1
Reviewer 1 Report
In this manuscript, the authors studied how cement paste penetrates a particle bed by carrying out a numerical simulation using a phase-field method combined with a continuous visco-plastic model. The impacts of three factors, i.e., average grain diameter, contact angle and rheological property of cement pastes have been investigated. In general, this manuscript is well organized, however, there are grammar and formatting errors throughout. Other concerns to the reviewer include:
- In terms of additive manufacturing of cementitious materials, the motivation of this study is not very clear. How the result of this research can benefit this additive manufacturing process?
- If the yield stresses keep going higher (> 4 Pa), how will the penetration depth change for θ = 5π/8, and D50 = 2.6 mm? Will the penetration depths converge to plateau values as well?
- The discrepancy between simulation and experimental results are not negligible, and the authors admitted that the proposed model was not able to capture the interactions at the fluid-solid interface. Then what are the merit and limitation of this study?
- Through the simulation, the authors concluded that the contact angle was the most significant factor. How does this conclusion help to improve the 3D printing of concrete?
Author Response
REVIEWER 1
In this manuscript, the authors studied how cement paste penetrates a particle bed by carrying out a numerical simulation using a phase-field method combined with a continuous visco-plastic model. The impacts of three factors, i.e., average grain diameter, contact angle and rheological property of cement pastes have been investigated. In general, this manuscript is well organized, however, there are grammar and formatting errors throughout. Other concerns to the reviewer include:
- In terms of additive manufacturing of cementitious materials, the motivation of this study is not very clear. How the result of this research can benefit this additive manufacturing process?
Thanks for this comment. This sentence has been added to the paper (L70):
The main challenge of this study is to add knowledge about the control of the paste intrusion into the particle bed layer. As the future strength and final shape accuracy is related to the degree of paste penetration, it is a major concern for additive manufacturing of cementitious materials by selective paste intrusion. To bond the individual layers, the cement paste should penetrate the particle bed layer over its entire thickness.
This sentence has been added to the paper (L70):
As the future strength and final shape accuracy is related to the degree of paste penetration, it is a major concern for additive manufacturing of cementitious materials by selective paste intrusion.
- If the yield stresses keep going higher (> 4 Pa), how will the penetration depth change for θ = 5π/8, and D50 = 2.6 mm? Will the penetration depths converge to plateau values as well?
Thanks for this question. This penetration depth will converge to a plateau value which is lower as the yield stress value rules the final stoppage of the cement paste. Higher values of yield stress will lead to uncomplete penetration even with a average grain diameter of 2.6 mm and a wettability characterized by θ = 5π/8.
- The discrepancy between simulation and experimental results are not negligible, and the authors admitted that the proposed model was not able to capture the interactions at the fluid-solid interface. Then what are the merit and limitation of this study?
The discrepancy could come from the different scales of the problem: the numerical modelling is a 2D axi-symmetric study, the analytical model is only considering 1D effect and the experiments are carried out in 3D. Thus, the merit is that the numerical model is enough to provide a quantitative view about the final penetration value. The limitation come from different simple assumptions made for the numerical model: no-absorption of the cement paste by the aggregates, no thixotropic properties of the cement paste, no drying effects in porous media, and others parameters that are unknown by the authors and that still needs further research.
- Through the simulation, the authors concluded that the contact angle was the most significant factor. How does this conclusion help to improve the 3D printing of concrete?
This sentence has been added (L317):
It then allows improving the penetration process through the use of surface tension agents that will the wettability characteristics.
Experiments with this kind of additives are on-going.
The author studied and compared the numerical modeling of the flow using a 2D axisymmetric with the experimental results. They evaluated penetration depth as a function of three variables in their study, including the average size of particles, the contact angle, and yield stress. In terms of writing, the author should be consistent in the use of the first-person point of view or third-person. The paper was well organized but lacked some justification of the results. It will be acceptable for publication with some revisions based on the comments.
Thanks for your comments that allowing improving the paper.

Reviewer 2 Report
The author studied and compared the numerical modeling of the flow using a 2D axisymmetric with the experimental results. They evaluated penetration depth as a function of three variables in their study, including the average size of particles, the contact angle, and yield stress. In terms of writing, the author should be consistent in the use of the first-person point of view or third-person. The paper was well organized but lacked some justification of the results. It will be acceptable for publication with some revisions based on the comments.
Reviewer comments:
The author studied and compared the numerical modeling of the flow using a 2D axisymmetric with the experimental results. They evaluated penetration depth as a function of three variables in their study, including the average size of particles, the contact angle, and yield stress. In terms of writing, the author should be consistent in the use of the first-person point of view or third-person. The paper was well organized but lacked some justification of the results. It will be acceptable for publication with some revisions based on the comments:
Introduction:
In general, the introduction is good and covered a brief summary of the literature, but it lacks the discussion of the application
It would be informative for the reader if the author can provide some description of the challenges on the commercialization of the particle bed 3D-printing? What would be the possible application for particle bed 3D-printing compared to extrusion-based techniques?
Material and Methods
Line 130, what is the void fraction?
Lines 132, 140, 145, 190, 205, 221, 229, 233, 258, 274, 285, 288,303, 306, the references are missed
In equation (1), the author should have introduced all of the variables, coefficients, and factors in the equation and explain how to find them from the shear stress-shear rate curve.
Line 161, change figure 3 to Figure 3.
In Figure 3. How author determined the experimental yield stress, according to the experimental curves for blue and red ones, the experimental yield stresses are similar, and they are not 2 Pa; similarly, from the Papanastasiou model, the yield stress of solid red line is 0.2 Pa and for the solid blue line is 0.1 Pa.
In Figure 3, it is not clear what is the difference between blue and red curves. Are they the experimental results of two different pastes? The author should clarify and explain well how they extracted different parameters from the curves through the manuscript since, in the current version, it is so confusing.
In Figure 3, there is a type in the figure for the presentation of shear rate symbol at the top of the curve where for shear rate equals 0.01 s-1
In equation (2), the author should have introduced all of the variables (including t)
Line 189: where are these equations from? Cite the proper references
Line 204-205: Valid references must be provided.
Result
Line 231, why the author just evaluates the effect of dave=1.6mm. Increasing the size of particles means a lower surface area, and it could change the results of the contact angle.
Line 232, why should the ratio be equal to the size of aggregates?
Figures 7 and 8, what does it mean when the curve is a plateau for the red curve. The author should explain it in the result
Figures 7 and 8, it is interesting that for contact angle=90 degree and accordingly, the capillary force ?, the trend of penetration curve changes and becomes somewhat similar to the red curve and plateau. The author should try to justify this observation. Why, by changing yield stress, it behaves similarly to when it tends to pull the paste upward in the blue curve.
Lines 258-260, what is the contact angle for this evaluation.
Line285-293, what was the void fraction of the experimental case (Refrence#27)
Line 320-321, it would be interesting if the author could add the effect of contact angle for different particle sizes in the plots
Figure 11, 12, why the author shows the standard deviation for the simulation results? What does even mean to have a variation in the results of the simulation!
Author Response
REVIEWER 2
Reviewer comments:
The author studied and compared the numerical modeling of the flow using a 2D axisymmetric with the experimental results. They evaluated penetration depth as a function of three variables in their study, including the average size of particles, the contact angle, and yield stress. In terms of writing, the author should be consistent in the use of the first-person point of view or third-person. The paper was well organized but lacked some justification of the results. It will be acceptable for publication with some revisions based on the comments:
The minor revisions have been carried out and completes answers to the reviewer comments are listed below.
Introduction:
In general, the introduction is good and covered a brief summary of the literature, but it lacks the discussion of the application
It would be informative for the reader if the author can provide some description of the challenges on the commercialization of the particle bed 3D-printing? What would be the possible application for particle bed 3D-printing compared to extrusion-based techniques?
Thanks for this comment. The adavantage was listed in L 65-57. To sump-up, The main advantage of the SPI method is its capability to produce high-resolution and free-form elements which permit news way for structural and architectural design.
This sentence has been added to the paper (L):
L67-68 This technique could also be resource-efficient as less polymers additives are needed in the final mix-design of the element.
Material and Methods
Line 130, what is the void fraction?
The void fraction is defined as the ratio of the volume of the air phase occupied in the aggregate layer.
Lines 132, 140, 145, 190, 205, 221, 229, 233, 258, 274, 285, 288,303, 306, the references are missed
We do not understand this comment as the references clearly appear in the submitted document. Exchanges with the editorial offices should help to fix this remark.
In equation (1), the author should have introduced all of the variables, coefficients, and factors in the equation and explain how to find them from the shear stress-shear rate curve.
Thanks for this comment
In Line 163-164, this sentence has been changed:
Values of experimental yield stress and viscosity are evaluated from the fitting of the Bingham model on the measured data from rheological measurements carried out in a previous work of Pierre et al [27] are used in this study and can be seen in figure 3
Line 161, change figure 3 to Figure 3.
Thanks, it has been corrected.
In Figure 3. How author determined the experimental yield stress, according to the experimental curves for blue and red ones, the experimental yield stresses are similar, and they are not 2 Pa; similarly, from the Papanastasiou model, the yield stress of solid red line is 0.2 Pa and for the solid blue line is 0.1 Pa.
Concerning the experimental data (empty squares in red and blue), as a log scale is used, the yield stress is 2 Pa for the bleu symbol and 4 Pa for the red symbol (plateau value at very low shear rate).
About the Papanastasiou model, an exponential stress-growth term (m) is proposed to render the original Bingham model as a one with high viscosity in the limit of low shear rates followed by a continuous transition to a viscous liquid. Thus, the value of 0.2 Pa and 0.1 Pa are not representative of the yield stress. Also as the shear rate imposed to the paste during the penetration does stay higher than 0.01 s-1, the Papanastasiou model is reliable (see the border unreliable-reliable domains in the Figure 3)
In Figure 3, it is not clear what is the difference between blue and red curves. Are they the experimental results of two different pastes? The author should clarify and explain well how they extracted different parameters from the curves through the manuscript since, in the current version, it is so confusing.
Thanks for this remark; we clarify with this new sentence:
Values of experimental yield stress and viscosity are evaluated from the fitting of the Bingham model on the measured data from rheological measurements on two different cement pastes carried out in a previous work of Pierre et al [27] are used in this study and can be seen in Figure 3.
In Figure 3, there is a type in the figure for the presentation of shear rate symbol at the top of the curve where for shear rate equals 0.01 s-1
Thanks for this observation; a dot has been added to refer to a shear rate and not a shear strain.
In equation (2), the author should have introduced all of the variables (including t)
Thanks for this comments.
The symbol t refers to the time and we add this in the text.
Line 189: where are these equations from? Cite the proper references
These equations could be found in reference 35 (it has been added in the text before the equations)
“Feng Bai, Xiaoming He, Xiaofeng Yang, Ran Zhou, Cheng Wang, Three dimensional phase-field investigation of droplet formation in microfluidic flow focusing devices with experimental validation, International Journal of Multiphase Flow, Volume 93, 2017, Pages 130-141, ISSN 0301-9322
Line 204-205: Valid references must be provided.
We do not understand this comment as the references clearly appear in the submitted document. Reference [28] is already in the text.
Result
Line 231, why the author just evaluates the effect of dave=1.6mm. Increasing the size of particles means a lower surface area, and it could change the results of the contact angle.
This value is chosen as it allows balancing the surface tension effect vs. the gravity effect. The computation of leads to a value of 1.6 mm; thus it is the most representative value to study the influence of the contact angle. Then, we did not judge that is useful to extent this study to the others grains diameters as the tendency is clear enough. Also, we agree that increasing the size of the particle will change the penetration depth results.
Line 232, why should the ratio be equal to the size of aggregates?
Thanks for this question.
As the surface tension effect value is 0.07 N/m , the density is 200 kg/m3 and the gravity is 9,81; a value of 1.8 mm. is computed for the equation ; then the closer value is the average grain diameter of 1.6 mm to study the influence of the contact angle. Also, we should expect the same tendency for the others average grains diameter.
Figures 7 and 8, what does it mean when the curve is a plateau for the red curve. The author should explain it in the result
A plateau value means that the penetration has reached its final value through the layer thickness.
Figures 7 and 8, it is interesting that for contact angle=90 degree and accordingly, the capillary force ?, the trend of penetration curve changes and becomes somewhat similar to the red curve and plateau. The author should try to justify this observation. Why, by changing yield stress, it behaves similarly to when it tends to pull the paste upward in the blue curve.
Thanks for this observation and remarks.
It seems that the pressure due to the surface tension effect could sometimes drive upward or downward the fluid as the radius of curvature of the grains change and the contact angle must stay close to a value of π/2. Thus the main point is to distinguish the yield stress effect regarding the same value of contact angle.
Lines 258-260, what is the contact angle for this evaluation.
We add this in the text but the information was already in L247-248:
Nevertheless, as we want to compare the numerical modeling results with previous existing results taking into account the rheological properties as well as the morphology and granular distribution of the granular-bed, we choose a value of the contact angle of π/2.
Line285-293, what was the void fraction of the experimental case (Refrence#27)
The void fraction was 0.46, as in the numerical generation of the particle bed.
Line 320-321, it would be interesting if the author could add the effect of contact angle for different particle sizes in the plots
The contact angle is here π/2. As we do not want to add to much information and to focus on the comparison of the experimental results, the analytical modelling results and the numerical modelling results, we would rather not add this in the figures.
We hope that the reviewer will understand our point of view.
Figure 11, 12, why the author shows the standard deviation for the simulation results? What does even mean to have a variation in the results of the simulation!
The error bars for the simulation results highlight the mesh size. As the mesh size used is relatively small, it then allows providing uncertainties on the final penetration depth evaluation.
